# The Association of Sodium or Potassium Intake Timing with Athens Insomnia Scale Scores: A Cross-Sectional Study

**DOI:** 10.3390/nu17010148

**Published:** 2024-12-31

**Authors:** Taiki Okamoto, Yun-Peng Lo, Inn-Kynn Khaing, Shoko Inoue, Ayako Tada, Mikiko Michie, Tatsuhiko Kubo, Shigenobu Shibata, Yu Tahara

**Affiliations:** 1Graduate School of Biomedical and Health Sciences, Hiroshima University, Hiroshima 734-0037, Japan; b204755@hiroshima-u.ac.jp (T.O.); m233079@hiroshima-u.ac.jp (Y.-P.L.); d226753@hiroshima-u.ac.jp (I.-K.K.); tkubo@hiroshima-u.ac.jp (T.K.); shibatas@hiroshima-u.ac.jp (S.S.); 2Asken Inc., Shinjuku-ku, Tokyo 163-1408, Japan; inoues@greenhouse.co.jp (S.I.); tadaay@greenhouse.co.jp (A.T.); mikiko-michie@greenhouse.co.jp (M.M.); 3School of Advanced Science and Engineering, Waseda University, Shinjuku-ku, Tokyo 162-0056, Japan

**Keywords:** sleep, nocturnal hypertension, potassium, sodium, circadian rhythm

## Abstract

Background/Objectives: Insomnia is a significant public health problem affecting a large population. Although previous research has explored the relationship between specific nutrients and insomnia, comprehensive analyses of daily eating patterns of macro- and micronutrients remain limited. Since nocturnal hypertension is related to sodium/potassium intake and sleep disturbances, the present cross-sectional study hypothesized that daily eating patterns of potassium and sodium would be associated with Athens Insomnia Scale (AIS) scores. Methods: Dietary data (breakfast, lunch, dinner, and snack intake) and additional online questionnaire responses were obtained from users (N = 4568; age: 43.5 ± 11.2 years) of Asken, a Japanese food-logging mobile health application. AIS scores were divided into three groups according to symptom classification. Results: Multiple regression analyses revealed that individuals with higher AIS scores had lower daily potassium intake; potassium at dinner was especially crucial. Conclusions: These results underscore the potential importance of potassium intake in relation to sleep and highlight a novel association between sleep disturbances and the timing of sodium and potassium consumption.

## 1. Introduction

Sleep disorders, such as insomnia, are major public health problems that affect a large population [1]. As insomnia impairs quality of life, increases the risk of other diseases, and causes an economic burden estimated to be tens of billions of dollars annually, it is important to identify the daily behaviors related to these sleep problems [2]. Previous studies have examined the relationship between macronutrient intake and sleep [3,4,5,6,7,8,9,10]. For instance, high-carbohydrate/low-fat diets or late-night carbohydrate intake were found to influence sleep architecture and quality, and specific amino acids like tryptophan, glycine, and l-serine are associated with improved sleep outcomes [3,4,5,6,7,8,9]. Meanwhile, Grandner et al. analyzed micronutrient intake related to sleep using data from the 2007–2008 U.S. National Health and Nutrition Examination Survey [10]. They found that difficulty in maintaining sleep was associated with nutrients such as salt, butanoic acid, carbohydrates, dodecanoic acid, vitamin D, lycopene, and hexanoic acid, while theobromine and potassium were linked to daytime sleepiness [10].

Nocturnal hypertension is associated with sleep disturbance. A meta-analysis showed that hypertension patients of the non-dipper type, whose blood pressure does not decrease at night, had poorer sleep quality scores compared with the dipper type [11]. It is also well-known that dietary sodium and potassium are related to blood pressure [12]. Dietary intake patterns high in sodium, low in potassium, or both were associated with altered nocturnal blood pressure decrease and the non-dipper type [13]. Additionally, in an intervention study, late sleep and mealtimes were associated with higher sodium intake than normal sleep [14]. Korean adolescents showed that the higher the sodium intake, the shorter the sleep duration [15]. Low potassium intake, indicated by decreased potassium excretion in the urine, was associated with poor sleep quality, especially in women [16]. A community-based cohort study of Hispanics aged 18–74 years in four US cities showed that short sleepers (≤6 h) had significantly lower potassium intake than others [17]. These findings highlight the potential importance of diet in regulating blood pressure, which may also be associated with sleep health.

As most previous studies used the total daily intake, they did not examine the relationship between sleep and nutrient data for each meal timing (e.g., breakfast and dinner). Furthermore, limited research focused on the dietary patterns of sodium/potassium, which are related to hypertension and sleep disturbances. Therefore, in the current study, we hypothesized that daily eating patterns of potassium and sodium would be associated with sleep disturbances.

## 2. Materials and Methods

### 2.1. Ethical Considerations

This study received approval from the Ethics Review Committee on Research with Human Subjects at Waseda University (no. 2021-101) and was conducted in accordance with the guidelines of the Declaration of Helsinki. It was designed, conducted, and analyzed following the STROBE statement. Informed consent was obtained from all participants upon initiation of app usage and completion of a web survey. Participants anonymously completed the questionnaire to protect their privacy and ensure confidentiality. The protocol has been registered at UMIN (ID: UMIN000043055). The data collection aimed to investigate the relationship between body weight and food intake as the primary investigation. The current analysis constitutes a secondary analysis.

### 2.2. Food-Logging Mobile Health App “Asken”

“Asken” is a widely used Japanese mobile health application for food logging and coaching, boasting over 11,242,000 downloads as of November 2024 [18]. Self-reported food logs accumulated in the app have been validated as reliable for research purposes. Users can input ingredients, dishes, and portion sizes into the app. The app automatically calculates calorie and nutritional intake from food records by referencing the *Standard Tables of Food Composition in Japan*, 2020 Edition (8th revision), as determined by the Ministry of Education, Culture, Sports, Science, and Technology. The app also provides feedback on the value of nutritional intake based on the Dietary Reference Intakes for the Japanese (2020), as determined by the Ministry of Health, Labor, and Welfare.

### 2.3. Participants and Data Inclusion and Exclusion

An online survey for this cross-sectional study was conducted among Asken users at the end of January 2021. The participant inclusion criteria were limited to users aged 20–64 years, without any specific disease requirements. From an initial pool of 7333 app users, we selected participants who recorded food logs for 10 or more days per month; reported their sex, weight, and body mass index (BMI); and were non-shift workers. Finally, 4568 participants were included in this study (3320 women and 1248 men).

### 2.4. Dietary Data

The average dietary data spanning one month (January–February 2021) from Asken were utilized. The current study used data on energy intake and intake of protein, fat, carbohydrates, sodium, and potassium during breakfast, lunch, dinner, and snacks. Missing values in the food logs were excluded from the average calculation because we could not verify whether they resulted from skipped meals or data omission.

### 2.5. Questionnaires

The questions included 3 items on basic characteristics (sex, age, and prefecture of residence), 2 items on eating habits, 6 items on physical activity (a short version of the International Physical Activity Questionnaire), 8 items on sleep disturbances (Athens Insomnia Scale (AIS) score), 10 items on sleep habits (including average sleep duration per week) and chronotype (a short version of the Munich Chronotype Questionnaire), 10 items on personality assessment (Ten-Item Personality Inventory), subjective well-being, health, and physical fitness.

### 2.6. Sleep Assessment

The sleep disturbances were assessed using the Athens Insomnia Scale [19], a self-administered psychometric questionnaire designed to evaluate sleep disorders, particularly insomnia [20]. It consists of eight items rated on a Likert scale ranging from 0 “no problem at all” to 3 “very severe” [20]. The total score ranges from 0 (absence of any sleep-related problems) to 24 (the most severe degree of insomnia). Severity is classified as normal for scores of 3 or less, subclinical insomnia for scores of greater than 3 but less than 6, and clinical insomnia for scores of 6 or more [19,21,22].

### 2.7. Statistical Analyses

Sample size analysis was conducted as described in previous papers using the same dataset [23]. All statistical analyses were performed using IBM SPSS Statistics (version 27.0; IBM Ltd., Armonk, NY, USA). Characteristics were presented as means and standard deviations (Table 1). Because most samples did not pass the Kolmogorov–Smirnov normality test, a nonparametric analysis was used in this study. The Kruskal–Wallis test was conducted to understand the general characteristics associated with AIS severity and eating behavior (Table 2). Furthermore, a logarithmic transformation was applied before conducting multiple linear regression analyses to evaluate the association between sodium and potassium intake and sleep disturbances while adjusting for potential confounders (Table 3). The dependent variable was the log-transformed AIS score, while the independent variables included sodium, potassium, and the sodium-to-potassium ratio. To control for potential confounders, the model was adjusted for sex, age, BMI, total daily energy intake, physical activity, mid-sleep on free days corrected for sleep debt accumulated during workdays (MSFsc), social jetlag (SJL), breakfast frequency, and night snack frequency. In all multiple regression analyses, the variance inflation factor (VIF) for each explanatory variable was below 10, indicating no evidence of multicollinearity. A value of *p* < 0.05 was considered a significant difference.

## 3. Results

### 3.1. Basic Characteristics

The general characteristics of the participants are presented in Table 1. Out of the 4568 participants, 72.7% were female. The characteristics of the participants classified according to AIS severity are shown in Table 2. Based on the Kruskal–Wallis test, significant differences in sleep disturbance severity were observed across several variables, including age (*p* < 0.001), physical activity (*p* < 0.001), total daily potassium intake (*p* < 0.001), and the sodium-to-potassium intake ratio (*p* = 0.004). Significant differences were also found in breakfast energy intake (*p* = 0.009), potassium intake (*p* < 0.001), and the sodium-to-potassium ratio (*p* = 0.018); lunch energy intake (*p* = 0.042) and potassium intake (*p* = 0.002); dinner carbohydrate intake (*p* = 0.006), potassium intake (*p* = 0.001), and the sodium-to-potassium ratio (*p* = 0.002); and snack energy intake (*p* < 0.001), sodium intake (*p* < 0.001), potassium intake (*p* = 0.027), and the sodium-to-potassium ratio (*p* < 0.001).

### 3.2. Association Between AIS Score and Dietary Patterns of Sodium and Potassium

Multiple regression analysis was conducted to investigate the association between AIS scores and dietary patterns of sodium and potassium intake (Table 3). Total daily potassium intake was inversely associated with log AIS score (β = −0.036; *p* = 0.034). When intake at each meal (breakfast, lunch, dinner, and snacks) was analyzed separately, only potassium intake at dinner remained significantly associated with AIS score (β = −0.066; *p* = 0.003), suggesting that higher potassium intake at dinner may be linked to fewer sleep disturbances. No significant associations were observed for the sodium-to-potassium ratio.

## 4. Discussion

In this study, we examined the relationship between potassium and sodium intake and AIS scores among Japanese adults. The findings suggest that potassium intake, rather than sodium, may be related to sleep, with higher total daily potassium intake—especially at dinner—being inversely associated with AIS scores, indicating fewer sleep disturbances. However, the sodium-to-potassium ratio showed no significant association with the severity of AIS scores.

The daily distributions of sodium and potassium intake in the present study cohort were consistent with the results of the Japanese National Health and Nutrition Survey [24]. As shown in Table 2, the highest sodium and potassium intake was observed at dinner, followed by lunch, breakfast, and snacks. In addition, sodium intake was higher than potassium intake in all meals, except snacks.

Multiple regression analysis indicated that individuals with a higher potassium intake had lower AIS scores. Previous studies have shown that lower potassium intake is associated with short sleepers, sleepiness during the day, poor sleep quality, and nighttime awakening and sleep efficiency [17,25]. However, these previous studies focused on the total daily intake of potassium without mentioning the intake timing. Our data indicated that potassium intake, especially at dinner, was negatively associated with AIS scores. There are several potential mechanisms explaining the relationship between potassium intake and sleep. Firstly, adequate potassium levels are critical for proper muscle contraction and relaxation, as potassium plays a key role in transmitting electrical signals in nerve fibers and muscle cells, thereby regulating overall muscle function and contributing to sleep patterns [26,27]. Secondly, potassium is essential for neurotransmitter function. Ion channels in the brain maintain neuronal excitability and have been identified as key regulators of sleep duration and the transition between wakefulness and sleep [26,28]. Cortical potassium channels modulate sleep oscillations, operating in a frequency-specific manner to regulate processes such as slow-wave activity, which is fundamental to sleep quality and restoration [28]. Lastly, increased potassium intake has been shown to be related to lower blood pressure [29,30]. Studies have demonstrated that potassium reduces both systolic and diastolic blood pressure through mechanisms such as promoting natriuresis and kaliuresis, modulating baroreceptor sensitivity, and attenuating vasoconstrictive responses to norepinephrine and angiotensin II [29,31,32]. Given the well-documented association between poor sleep and hypertension, these findings suggest that potassium may be related to better sleep by regulating blood pressure. However, the mechanism underlying why dinner intake is particularly significant remains to be investigated in future studies.

Several studies have shown that sodium or salt intake is associated with sleep maintenance difficulties [10], later sleep times [14], and a higher probability of insomnia [33]. A Japanese cross-sectional study showed that energy-adjusted sodium intake was significantly positively correlated with sleep duration among healthy Japanese adult men [34]. However, we did not find any significance between sodium intake and AIS scores in the current study. These findings suggest that further evidence is required to verify the relationship between sodium intake and sleep.

The Japanese diet has been characterized by high sodium and low potassium intake, contributing to a high sodium-to-potassium ratio [35]. The sodium-to-potassium ratio serves as a superior indicator for assessing blood pressure outcomes and hypertension incidence compared with individual measures of sodium or potassium. The sodium–chloride cotransporter (NCC), which is expressed in the distal convoluted tubules of the kidneys, is a key molecule in the regulation of urinary potassium excretion [36]. Consuming a low-potassium diet activates the NCC, promoting sodium reabsorption. Conversely, a high-potassium diet strongly suppresses NCC activation, resulting in increased urinary sodium and potassium excretion [36]. However, the current results did not find any significant association between the sodium-to-potassium intake ratio and AIS scores.

This study had some limitations. First, dietary data were self-reported, which may have introduced response bias. Additionally, self-reported sleep symptoms are nonspecific and may reflect various underlying causes, including specific sleep disorders such as insomnia and sleep apnea. It is possible that the nutrient distribution observed in this study differs from the general population, as 95% of the app users aimed to lose weight, and 70% were female. Furthermore, specific mealtimes were not recorded; thus, we could not estimate whether the snacks were consumed as nighttime meals or during the daytime. While arterial blood pressure may be associated with sleep, these data were not collected in this study. Finally, the cross-sectional study design limited the determination of causality.

## 5. Conclusions

Among healthy Japanese adults, this study identified a potential association between sleep problems and lower potassium intake, particularly at dinner. These findings suggest that dietary potassium may be related to sleep quality and provide a foundation for future intervention studies and personalized dietary recommendations. To better understand the observed association between potassium intake and insomnia, future research should include both dietary records and blood pressure data. Longitudinal and intervention studies will also be essential to establish causal relationships and explore other factors influencing sleep.

## Figures and Tables

**Table 1 nutrients-17-00148-t001:** General characteristics of participants.

	Overall	
	(*n* = 4568)	
Characteristics	Number (%)	
Sex		
Male	1248 (27.3)	
Female	3320 (72.7)	
	Mean	SD
Age (years old)	43.5	11.2
BMI (kg/m^2^)	23.4	4.1
AIS score	4.3	3.3
Physical activity (MET-minutes/week)	32.0	39.8
MSFsc (hh:mm)	03:36	01:16
SJL (hh:mm)	00:47	00:41
Breakfast frequency (times/week)	6.1	1.9
Night snack frequency (times/week)	2.0	2.7

Abbreviations: AIS, Athens Insomnia Scale; BMI, body mass index; SD, standard deviation; MSFsc, mid-sleep on free days corrected for sleep debt accumulated during workdays; SJL, social jetlag.

**Table 2 nutrients-17-00148-t002:** Characteristics of participants classified by AIS severity.

		Overall		Normal		Subclinical Insomnia		Clinical Insomnia		Kruskal–Wallis Test
				AIS ≤ 3		3 < AIS < 6		AIS ≥ 6		
		(*n* = 4568)		(*n* = 2194)		(*n* = 1159)		(*n* = 1215)		
		Mean	SD	Mean	SD	Mean	SD	Mean	SD	*p*-Value
Age (years old)		43.5	11.2	44.3	11.2	43.2	11.3	42.5	11.0	**<0.001**
BMI (kg/m^2^)		23.4	4.1	23.3	3.9	23.5	4.2	23.6	4.4	0.129
Physical activity (MET-minutes/week)		32.0	39.8	34.2	39.9	29.9	41.4	30.1	38.0	**<0.001**
Daily total	Energy (kcal/day)	1767.6	366.1	1772.8	366.4	1770.3	351.4	1755.6	379.3	0.253
	Protein (%/energy/day)	17.2	4.5	17.2	4.5	17.0	4.4	17.2	4.7	0.293
	Lipid (%/energy/day)	32.1	7.1	31.9	6.6	32.1	6.6	32.3	8.2	0.523
	Carbohydrates (%/energy/day)	48.9	10.3	48.6	11.4	49.0	8.8	49.3	9.2	0.147
	Sodium (mg/day)	3575.3	905.1	3598.5	899.6	3568.9	893.6	3539.6	925.2	0.203
	Potassium (mg/day)	2523.2	610.9	2562.4	614.8	2510.7	587.5	2464.5	621.0	**<0.001**
	Sodium-to-potassium ratio	1.6	4.5	1.5	2.2	1.7	8.4	1.5	0.5	**0.004**
Breakfast	Energy (kcal/meal)	374.2	137.0	380.1	136.6	369.4	133.4	368.2	140.8	**0.009**
	Protein (%/energy/meal)	17.4	7.3	17.4	7.2	17.1	6.7	17.6	7.9	0.746
	Lipid (%/energy/meal)	28.1	8.2	28.3	8.2	28.1	8.3	27.9	7.9	0.916
	Carbohydrates (%/energy/meal)	54.4	11.0	54.2	11.0	54.7	11.3	54.6	10.5	0.967
	Sodium (mg/meal)	658.4	373.8	666.1	370.7	646.0	370.6	656.4	382.2	0.217
	Potassium (mg/meal)	604.5	295.3	623.0	300.0	591.8	280.8	583.0	298.3	**<0.001**
	Sodium-to-potassium ratio	1.2	0.8	1.2	0.8	1.2	0.8	1.3	0.9	**0.018**
Lunch	Energy (kcal/meal)	552.1	137.9	554.8	137.5	556.8	138.2	542.9	138.1	**0.042**
	Protein (%/energy/meal)	16.8	4.1	16.9	4.0	16.7	4.1	16.7	4.1	0.149
	Lipid (%/energy/meal)	30.7	5.9	30.6	5.9	30.7	6.1	30.8	5.6	0.570
	Carbohydrates (%/energy/meal)	50.8	7.6	50.7	7.8	50.8	7.8	51.0	7.1	0.623
	Sodium (mg/meal)	1298.4	423.5	1307.0	415.1	1305.4	432.2	1276.1	429.8	0.113
	Potassium (mg/meal)	716.0	231.3	723.4	228.4	723.4	232.3	695.7	234.5	**0.002**
	Sodium-to-potassium ratio	1.9	0.7	1.9	0.7	1.9	0.8	1.9	0.7	0.284
Dinner	Energy (kcal/meal)	631.7	199.4	632.3	200.7	631.7	191.7	630.7	204.3	0.905
	Protein (%/energy/meal)	19.1	4.9	19.2	4.7	19.0	4.9	19.1	5.0	0.495
	Lipid (%/energy/meal)	33.3	6.5	33.4	6.6	33.5	6.5	33.1	6.3	0.359
	Carbohydrates (%/energy/meal)	42.1	9.4	41.7	9.3	42.2	9.7	42.9	9.1	**0.006**
	Sodium (mg/meal)	1474.4	487.5	1483.6	482.0	1470.9	480.1	1460.8	504.2	0.380
	Potassium (mg/meal)	993.3	299.4	1009.9	298.7	986.5	290.9	970.1	307.0	**0.001**
	Sodium-to-potassium ratio	1.5	0.5	1.5	0.5	1.5	0.5	1.6	0.7	**0.002**
Snacks	Energy (kcal/meal)	232.3	109.6	226.9	109.1	235.9	109.5	238.8	110.2	**<0.001**
	Protein (%/energy/meal)	12.3	8.1	12.5	8.3	12.1	8.2	12.1	7.7	0.164
	Lipid (%/energy/meal)	34.4	10.9	34.4	10.9	34.5	11.3	34.3	10.4	0.854
	Carbohydrate (%/energy/meal)	52.3	14.2	52.1	15.4	52.1	13.0	52.9	13.0	0.475
	Sodium (mg/meal)	178.7	136.5	170.9	131.3	177.6	132.9	193.9	147.4	**<0.001**
	Potassium (mg/meal)	250.9	150.6	250.4	151.4	245.2	151.3	257.1	148.2	**0.027**
	Sodium-to-potassium ratio	1.0	3.3	0.9	2.0	1.1	5.9	0.9	1.0	**<0.001**

Abbreviations: AIS, Athens Insomnia Scale; BMI, body mass index; SD, standard deviation; Significant *p*-values of the independent variable are presented in bold (*p* < 0.05).

**Table 3 nutrients-17-00148-t003:** Association between AIS score and sodium and potassium intake by multiple regression analysis.

Independent Variable		Dependent Variable: Log AIS Score					
		R^2^	B	β	95% CI		*p*-Value
					Lower	Upper	
Daily total	Sodium	0.021	0.000	−0.018	0.000	0.000	0.365
	Potassium		0.000	**−0.036**	0.000	0.000	**0.034**
Sodium	Breakfast	0.026	0.000	0.008	0.000	0.000	0.677
	Lunch		0.000	−0.036	0.000	0.000	0.091
	Dinner		0.000	0.012	0.000	0.000	0.610
	Snacks		0.000	0.034	0.000	0.000	0.075
Potassium	Breakfast		0.000	0.007	0.000	0.000	0.739
	Lunch		0.000	−0.011	0.000	0.000	0.563
	Dinner		0.000	**−0.066**	0.000	0.000	**0.003**
	Snacks		0.000	0.002	0.000	0.000	0.924
Sodium-to-potassium ratio	Daily total	0.019	0.001	0.005	−0.004	0.006	0.731
	Breakfast	0.021	−0.002	−0.003	−0.031	0.027	0.886
	Lunch		−0.005	−0.005	−0.040	0.030	0.784
	Dinner		0.048	0.035	−0.001	0.098	0.055
	Snacks		−0.001	−0.007	−0.008	0.005	0.659

Abbreviations: AIS, Athens Insomnia Scale; R^2^, adjusted coefficient of determination; B, partial regression coefficient; β, standardized partial regression coefficient; CI, confidence interval. Adjusted for sex, age, BMI, total daily energy intake, physical activity, MSFsc, SJL, breakfast frequency, and night snack frequency. Significant *p*-values of the independent variable are presented in bold (*p* < 0.05).

## Data Availability

The datasets presented in this article are not readily available because the data are part of an ongoing analysis. Requests to access the datasets should be directed to the corresponding author, Y.T.

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
