# Peer review of "The Association of Sodium or Potassium Intake Timing with Athens Insomnia Scale Scores: A Cross-Sectional Study"

_nutrients, 2024, doi:10.3390/nu17010148_

Round 1

Reviewer 1 Report

Comments and Suggestions for Authors

This is cross-section study based on food logs (application) on dietary intake timing and insomnia. Results are clearly presented and analyzed.

Minor comments:
Are there data on arterial blood pressure values?  If yes, it would be very important to include them in analysis, because insomnia is related to hypertension and hypertension is related to potassium /sodium intake.

Also, there is no analysis of physical activity and correlation with insomnia data. Please include that analysis in revised version of the manuscript because that can influence conclusions.

Author Response

This is cross-section study based on food logs (application) on dietary intake timing and insomnia. Results are clearly presented and analyzed.

Minor comments:

Are there data on arterial blood pressure values? If yes, it would be very important to include them in analysis, because insomnia is related to hypertension and hypertension is related to potassium /sodium intake.

Thank you for your valuable suggestion. Unfortunately, we did not collect arterial blood pressure in this study. We acknowledge the importance of this variable and suggest future studies to be included it to explore its relationship with insomnia and potassium/sodium intake.

Also, there is no analysis of physical activity and correlation with insomnia data. Please include that analysis in revised version of the manuscript because that can influence conclusions.

Thank you for highlighting this point. We have added a comparison of physical activity levels among different AIS groups (Table 2). The results show a significant difference in physical activity levels across groups (p<0.001).

Reviewer 2 Report

Comments and Suggestions for Authors

The authors should specify the type of regression analysis conducted (e.g., linear or logistic regression).

Additionally, they should report the assumptions underlying the multiple regression model used in their analysis.

The authors are also requested to provide the 95% confidence intervals for the regression models.

The first paragraph of the discussion section should present the main findings of the article rather than comparisons with other studies.

The authors should clearly describe how missing data were handled in their analysis.

Regarding the dietary data, it is unclear whether participants recorded their daily food intake using the app or if another methodology was employed. This lack of description should be addressed and clarified.

Comments on the Quality of English Language

Moderate English editing is required.

Author Response

The authors should specify the type of regression analysis conducted (e.g., linear or logistic regression).

Additionally, they should report the assumptions underlying the multiple regression model used in their analysis.

Thank you for the comment. We have clarified that linear regression analysis was used in this study. We also included an assessment of the assumptions for the multiple regression models checking the variance inflation factor (VIF).

The authors are also requested to provide the 95% confidence intervals for the regression models.

We have added the 95% confidence intervals for the regression models in the revised manuscript (Table 3).

The first paragraph of the discussion section should present the main findings of the article rather than comparisons with other studies.

Thank you for this suggestion. We have revised the first paragraph of the discussion section to highlight the main findings of this study.

The authors should clearly describe how missing data were handled in their analysis.

Thank you for comments. Missing data for sleep and demographic variables were excluded at the initial stage of analysis. For dietary records, as noted in section 2.4, missing values in the food logs were excluded from the average calculation because it was not possible to identify the reason either skipped meals or data omission.

Regarding the dietary data, it is unclear whether participants recorded their daily food intake using the app or if another methodology was employed. This lack of description should be addressed and clarified.

We appreciate this comment. Participants recorded their daily food intake using the Asken app. It was stated in Section 2.2 and 2.4. We have revised more clearly in the manuscript that dietary data used data from the app.

Reviewer 3 Report

Comments and Suggestions for Authors

This is an interestimg research article with adequate novelty. However, some points should be addressed.

- In the Introduction section, the sentence  "Previous studies have examined the relationship between macronutrient intake and sleep [3-10]." needs more analysis to obtain a more detailed overview of the topic of the article.

- The authors should ad references concerning the section 2.5 for Athens Insomnia Scale (AIS) score.

- In the statistical analysis section, the authors should report the normality test that they used for continious variables.

- In the section 3.1, the authors should add p-values for the significant associations into the text of the results.

- More text should be added in the Conclusions section.

Author Response

This is an interestimg research article with adequate novelty. However, some points should be addressed.

- In the Introduction section, the sentence “Previous studies have examined the relationship between macronutrient intake and sleep [3-10].” needs more analysis to obtain a more detailed overview of the topic of the article.

Thank you for this suggestion. We have expanded this section to include more detailed context.

- The authors should ad references concerning the section 2.5 for Athens Insomnia Scale (AIS) score.

We have added the appropriate references for the Athens Insomnia Scale (AIS) in section 2.6.

- In the statistical analysis section, the authors should report the normality test that they used for continious variables.

Thank you for this recommendation. We have included the normality test used for continuous variables in the statistical analysis section.

- In the section 3.1, the authors should add p-values for the significant associations into the text of the results.

We have added the p-values for significant associations into the results section as requested.

- More text should be added in the Conclusions section.

Thank you for your comment. We have revised and expanded the Conclusions section to provide more comprehensive insights and discuss future directions.

Round 2

Reviewer 2 Report

Comments and Suggestions for Authors

The authors have addressed all my comments

Author Response

Thank you for your review.

Reviewer 3 Report

Comments and Suggestions for Authors

The authors have significantly improved their manuscript.

Author Response

Thank you for your review.